# Compliance with vessel speed restrictions to protect North Atlantic right whales

Gregory K. Silber[1], Jeffrey D. Adams[1] and Christopher J. Fonnesbeck[2]

[1] Office of Protected Resources, National Marine Fisheries Service, NOAA, Silver Spring, MD, USA
[2] Department of Biostatistics, Vanderbilt University School of Medicine, Nashville, TN, USA

## ABSTRACT

Environmental regulations can only be effective if they are adhered to, but the motivations for regulatory compliance are not always clear. We assessed vessel operator compliance with a December 2008 regulation aimed at reducing collisions with the endangered North Atlantic right whale that requires vessels 65 feet or greater in length to travel at speeds of 10 knots or less at prescribed times and locations along the U.S. eastern seaboard. Extensive outreach efforts were undertaken to notify affected entities both before and after the regulation went into effect. Vessel speeds of 201,862 trips made between November 2008 and August 2013 by 8,009 individual vessels were quantified remotely, constituting a nearly complete census of transits made by the regulated population. Of these, 437 vessels (or their parent companies), some of whom had been observed exceeding the speed limit, were contacted through one of four non-punitive information programs. A fraction ($n = 26$ vessels/companies) received citations and fines. Despite the efforts to inform mariners, initial compliance was low ($<$5% of the trips were completely $<$10 knots) but improved in the latter part of the study. Each notification/enforcement program improved compliance to some degree and some may have influenced compliance across the entire regulated community. Citations/fines appeared to have the greatest influence on improving compliance in notified vessels/companies, followed in order of effectiveness by enforcement-office information letters, monthly summaries of vessel operations, and direct at-sea radio contact. Trips by cargo vessels exhibited the greatest change in behavior followed by tanker and passenger vessels. These results have application to other regulatory systems, especially where remote monitoring is feasible, and any setting where regulatory compliance is sought.

## INTRODUCTION

Natural resource conservation and management can take numerous forms, including through environmental regulations. However, environmental regulations are only effective if they are adhered to. A substantial body of socio-legal and economic literature has been devoted to the subject of regulatory compliance, but the factors that motivate individuals

Corresponding author
Gregory K. Silber,
greg.silber@noaa.gov

and businesses to comply are not always clear (*Gunningham & Kagan, 2005*; *May, 2005*). Compliance case studies have involved industrial pollution (*Kagan, Gunningham & Thornton, 2011*), hazardous waste (*Stafford, 2012*), agricultural practices (*Winter & May, 2001*), forestry (*Purdy, 2010*; *Peterson & Diss-Torrance, 2012*), fisheries (*Hønneland, 1999*; *Ali & Abdullah, 2010*), and endangered species conservation (*Langpap, 2006*; *Innes & Frisvold, 2009*), among others.

Some studies concluded that regulated communities may lack an understanding of the requirements or may lack the willingness or capacity to comply (*Burby & Paterson, 1993*; *Brehm & Hamilton, 1996*); others found that regulated entities may avoid complying because the consequences of noncompliance (i.e., enforcement actions) rarely outweigh the economic benefits of business as usual (*Winter & May, 2001*; *Tyler, 2006*). However, in many regulatory settings, limited resources may restrict enforcement actions and assessments of compliance to infrequent inspections (e.g., site visits), surveys, interviews, or self-reporting (*Winter & May, 2001*; *Gunningham, Kagan & Thornton, 2004*; *Gray & Shimshack, 2011*).

With regard to living marine resources, including endangered large whale protective measures, risk assessment estimates have been conducted (*van der Hoop, Vanderlaan & Taggart, 2012*; *Redfern et al., 2013*). But, there is also a need to ensure large whale conservation regulations are meeting their objectives through compliance.

## The problem of vessel collisions with large whales

Hundreds of fatal vessel collisions (or "strikes") with large whales have been reported, worldwide (*Laist et al., 2001*; *Van Waerebeek et al., 2007*). In fact, the actual number of strikes is likely far greater than the reported number because many go undetected or unreported. Collisions with ships are a serious threat to the recovery of the highly depleted North Atlantic right whale (*Eubalaena glacialis*) (*Kraus et al., 2005*) and collisions along with incidental entanglement in commercial fishing gear, have retarded the recovery of this species (*NMFS, 2005*). A link has been established between vessel speed and the likelihood of death of a vessel-struck whale whereby the probability of death of a whale involved in a collision increases as vessel speed increases (*Vanderlaan & Taggart, 2007*; *Conn & Silber, 2013*).

To address the threat to the recovery of the North Atlantic right whale, the National Oceanic and Atmospheric Administration's (NOAA) National Marine Fisheries Service (NMFS) issued regulations in November 2008 requiring all vessels 65 feet (19.8 m) and greater in length to travel at 10 knots or less in areas where North Atlantic right whales and high vessel density co-occur (*NMFS, 2008*). These areas, called seasonal management areas (SMA), are located along the east coast of the U.S. Atlantic seaboard and are active for fixed periods of the year that correspond with seasonal North Atlantic right whale migration, feeding, calving and nursery activities (Fig. 1). The regulations are broad in geographic scope and affect a substantial number of entities, including nearly all tanker, cargo (e.g., container ships, vehicle transport vessels), passenger vessels, and ferries engaged in international and domestic transport of goods and people entering major U.S. ports.

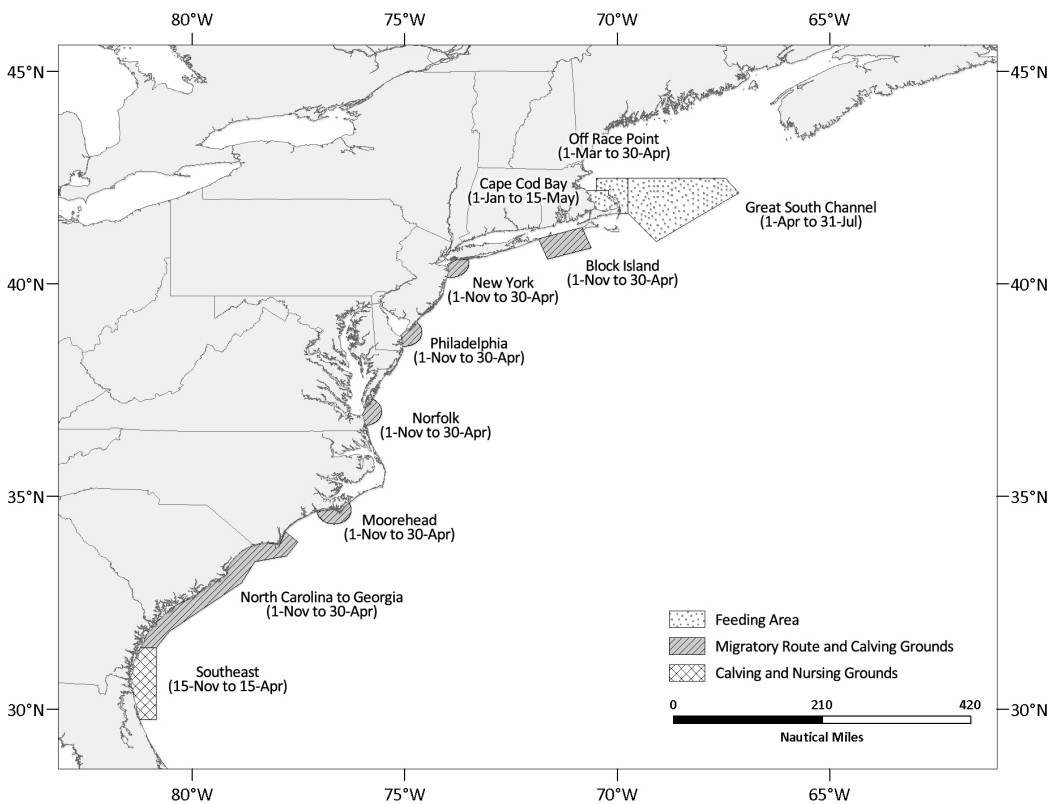

**Figure 1** Map depicting the location and active periods of the north Atlantic right whale seasonal management areas (SMAs).

## Notifying the affected community

Extensive efforts by a number of agencies were made—both prior to the regulations going into effect and on an ongoing basis while they were in effect—to notify the affected community about the speed regulations that included an array of broadcast, print, and electronic media outlets (Appendix S1; *Silber & Bettridge, 2012*). Knowledge of, and adherence to, the requirements, precautions, and safety-at-sea provisions contained in a number of the print and broadcast notification outlets (e.g., U.S. Coast Pilot publications, Broadcast Notice to Mariners) is mandatory for any vessel sailing in U.S. waters. Most vessels studied here are engaged in regular and periodic domestic and international routes that would have resulted in repeated exposure to notification about the speed regulations. Given the breadth of the notification efforts, we believe vessel operators should have had ample knowledge of the requirements.

## Compliance information and enforcement programs

After the restrictions went into effect, a subset of the regulated vessel operators or their companies received notifications and/or citations/fines under one of the information and/or enforcement programs described below when violations of the rule were detected. The programs were independently developed and carried out by four federal entities: two

by NOAA's Offices of General Counsel and Law Enforcement, and one each by the United States Coast Guard (USCG) and NMFS's Office of Protected Resources (OPR). There was no standardization or coordination between programs regarding protocols for notifying particular vessels/companies or the identity of operators being contacted. Each of these programs/activities is described immediately below in the chronological order in which they were first implemented.

### Hailing at-sea

In four periods during the first five years of the regulations (February–May 2009, January–July 2010, November 2010–July 2011, and January–March 2012), USCG personnel radioed vessels that were observed (detected via radar, Automatic Identification System (AIS), and/or visually) violating the speed restrictions and requested that the vessels slow to appropriate speeds. It was the only program involving real-time, verbal notification. It was also somewhat limited in scope, having been conducted in only six of 10 SMAs (the Great South Channel, Race Point, Cape Cod Bay, Philadelphia, Norfolk, and North Carolina to Georgia) and only when USCG cutters were on routine patrols or engaged in other missions.

### Community oriented policing and problem solving (COPPS) letters

As part of its Community Oriented Policing and Problem Solving (COPPS) Program, NOAA's Office of Law Enforcement (OLE) sent a total of 85 letters between September 2009 and January 2010 to companies whose vessel operators were determined by OLE agents (based on AIS data analysis) to have made at least one trip in an SMA that far exceeded the 10-knot limit. The letters were informative rather than punitive, and included detailed information regarding the observed violation(s) and a reminder about the speed restrictions.

### Notice of violation and assessment of civil penalties (NOVA)

To prosecute violations of the Endangered Species Act, NOAA's Office of General Counsel Enforcement Section can issue a Notice of Violation and Assessment of civil penalties (NOVA). A NOVA charges the respondent with a violation of laws and regulations, and assesses a civil monetary penalty in accordance with the agency's penalty policy for that violation (http://www.gc.noaa.gov/documents/031611_penalty_policy.pdf). Limited staff time required that attention be focused on a small number of vessels that exhibited numerous and flagrant breaches of the speed restrictions (as indicated by AIS), even though hundreds of violations were observed. Multiple offending trips were often cited in the NOVAs and fines were cumulative. Depending on the number of violations, penalties ranged from $5,750 to $92,000 (mean = $21,845) (www.gc.noaa.gov/enforce-office3. html). A total of 28 NOVAs were issued between November 2010 and September 2012 (and used to examine recipients' operations described below): seven in November 2010; two in December 2010 (those issued in November and December 2010 were defined as "season 3" for our purposes); eight in November 2011 (season 4); one in July 2012; three in August 2012; and seven in September 2012 (these latter three collectively were considered season 5). NOVAs issued in 2013 were not included in this analysis.

*Monthly summaries of vessel operations*

In collaboration with the World Shipping Council (WSC) and Chamber of Shipping of America (CSA), two industry trade associations that represent more than 90% of the world's international commercial shipping fleet, NMFS's OPR developed a program to disseminate AIS-based vessel operations information to WSC and CSA member companies. A total of 17 shipping companies (13 WSC and 4 CSA members; ca. 400 vessels) participated in the program. Starting in December 2010, and monthly for the duration of the study, OPR sent reports directly to company officials containing spreadsheet summaries of every vessel transit within active SMAs (regardless of whether the trip was compliant with the regulation) during the previous month which included: vessel name; date/time of entry into the SMA; distance traveled within the SMA; speeds when entering and exiting the SMA; and the mean and maximum speeds within the SMA.

### Study objectives

We sought to assess compliance by the regulated community by examining the response to the vessel speed restrictions. Using a remote monitoring program that provided a near-complete census of vessel operations, we quantified vessel operations in SMAs during the first five years of the regulations. In addition to quantifying overall compliance with the regulations, we assessed whether compliance with the regulations changed over time and whether attempts to improve compliance through the targeted notification and enforcement programs produced a change in behavior.

## MATERIALS AND METHODS

### Monitoring vessel operations

We examined vessel behavior using AIS data. AIS is a navigational safety system that transmits very high frequency (VHF) radio signals several times each minute. Each transmission contains static information specific to a given vessel which allowed us to assess compliance by individual vessels and more generally by principal vessel types. The signal also includes dynamic Global Positioning System-linked data unique to a particular voyage including location, heading, and speed (*Aarsæther & Moan, 2009*). Functioning AIS capabilities are required by the International Maritime Organization on all vessels ≥300 gross tons, and the USCG requires AIS on nearly all vessels sailing in U.S. waters. The USCG has established a national network of AIS receivers that provides coverage of nearly all U.S. coastal waters, a continuously sampled record of operations and, for us, a nearly complete census of the community subject to the speed limits. The AIS's reporting rates provided hundreds of records per trip and resulted in a large and rather precise record of vessel speed and operations.

### Assessing compliance

Using AIS data collected between November 1, 2008 and August 1, 2013, we analyzed all trips by vessels ≥65 feet in length that were located within the geographic boundaries of the SMAs (our analytical approach is described further in *Silber & Bettridge, 2010*). A trip located in an active SMA was considered compliant if all speeds were ≤10 knots. Because

binning trips as compliant/noncompliant in this way may not fully capture more subtle responses to the regulations (e.g., vessel operators who were not fully compliant but may have modified their behavior when travelling through active SMAs), we also calculated the percent of total transit distance traveled within SMAs at speeds >10 knots (PDGT10), and average speeds when all or a portion of the trip exceeded 10 knots. With the exception of the average speeds >10 knots metric, we did not calculate mean trip speeds because AIS signals are transmitted at regular and frequent time intervals and, as such, slower speeds are more heavily represented than higher speeds. PDGT10 is not influenced by the distributions of speed values, provides a standard measure that is independent of trip length or duration, and, along with average noncompliant speeds, allowed us to quantify degrees of compliance (or noncompliance).

The above-mentioned metrics (compliance, PDGT10, and average noncompliant speed) were quantified for vessels by type (vessel type analyses were limited to those principally impacted by the regulations, which included cargo, tanker, and passenger), by association with the different notification/enforcement programs (USCG Hailing At-Sea, COPPs Letters, NOVA, WSC and CSA Monthly Summaries), before and after they had received these notification/enforcement actions and for periods when the restrictions were not in effect. Summary statistics were generated for each SMA active season, which we define as beginning on the first day of November (coinciding with the opening of the migratory and calving grounds SMAs) and ending on July 31 of the following year (closing of the Great South Channel SMA) (Fig. 1).

## Statistical modeling

The observational design of the study made it difficult to directly associate changes in vessel behavior with any particular notification/enforcement program. The implementation of the suite of notification programs overlapped, confounding attempts to directly implicate any one action in the reduction of vessel speed. As such, we were limited to presenting summary statistics for the vessels associated with each notification/enforcement program.

We were, however, able to estimate the change in PDGT10 over time by examining the differences in its mean value across the sequence of the SMA active seasons during the first five years of the speed restrictions. A natural statistical model to describe the distribution of these values in a given season is the beta distribution, which is typically modeled as a function of scale ($\alpha$) and shape ($\beta$) parameters:

$$f(x \mid \alpha, \beta) = \frac{\Gamma(\alpha + \beta)}{\Gamma(\alpha).\Gamma(\beta)} x^{\alpha-1}(1 - x)^{\beta-1}.$$

We were interested in modeling the mean of this distribution (rather than the scale and shape specified above), so we reparameterized the beta distribution in terms of a mean $\mu$ and parameter $\nu$, which we interpret informally as a "sample size". Here we used the scaled distance of each segment as this sample size parameter, so that segments are weighted according to their length; we included the scale parameter as an unknown in the model, by

giving it a diffuse prior distribution. This reparameterization is:

$$\alpha = \mu \nu$$
$$\beta = (1 - \mu)\nu.$$

We expected the mean PDGT10 to vary with several factors, including three variables of interest: SMA, vessel type, and season. Thus, we modeled $\mu$ using a mixed effects model:

$$\mu_{ijk} = \theta_i + \psi_j + \phi_{ik}$$

where $\theta_i$ is the mean for vessel type $i$, $\psi_j$ is a random effect corresponding to SMA $j$, and $\phi_{ik}$ is the fixed effect of season $k$ on vessel type $i$. The first season in any SMA (either 2008 or 2009, depending on the SMA's location) is treated as the baseline; hence $\theta$ can be interpreted as the mean in the first season, and $\phi$ the effect of a subsequent season, relative to the first. It is these seasonal difference effects that are of primary interest. The random effect $\psi_j$ where $j = 1\ldots, S$ is modeled as:

$$\psi_j \sim N(0, \tau_\psi).$$

To account for individual variation not attributable to vessel type, season or SMA, we also employed a random effect, which draws a $\theta$ value from a normal distribution for each unique vessel.

For each scale parameter in the model ($\tau_\psi$, $\tau_\theta$, $\tau_\upsilon$), we specified a half-Cauchy distribution in the inverse square-root of the parameter:

$$f(\sigma \mid \beta) = \frac{2}{\pi \beta [1 + (\frac{\sigma}{\beta})^2]}.$$

This results in a relatively diffuse, weakly-informative prior (after transforming by $\tau = \sigma^{-2}$) that is easily overwhelmed by the data (*Gelman, 2006*).

Because typical at-sea speeds vary widely for the different vessel types, models were fit for each of the three most common vessel types in the dataset: cargo, tanker and passenger vessels. Model parameters were estimated using Markov chain Monte Carlo (MCMC) methods as provided by the PyMC (version 2.3, *Patil, Huard & Fonnesbeck, 2010*) software package. Each model was run for 20,000 iterations, with the first 10,000 conservatively discarded as burn-in, leaving 10,000 samples for inference. Models were checked for lack of convergence using the Gelman–Rubin statistic (*Gelman & Rubin, 1992*) and for lack of fit using posterior predictive checks (*Gelman et al., 2003*).

Our Bayesian logistic mixed-effects model generated estimates of the differences among seasons for different vessel types across all SMAs, along with corresponding measures of uncertainty, 95% posterior credible intervals. Intervals that include zero may be interpreted as not statistically different from zero. Interpreting coefficients on the inverse-logit scale is challenging, since the underlying function is non-linear. For a given parameter value, the effect will be larger near the middle of the logistic curve (0.5), where it is steepest, and smaller near the boundaries (0 and 1), where it is flat. Thus, it

**Table 1 Compliance metric summary statistics for trips through the SMAs during active and inactive periods by all vessels (cargo, tanker, and passenger) for the first five years of the speed restrictions.**

| Season | SMA status | Trips | Vessels | Compliance[a] | PDGT10 | Mean noncompliant speed[a] |
|--------|-----------|-------|---------|------------|--------|----------------------------|
| 1 | Active | 14907 | 1776 | 4.0 | 57.3 | 12.0 |
|   | Inactive | 25974 | 2401 | 1.7 | 83.4 | 14.3 |
| 2 | Active | 19439 | 2019 | 4.2 | 55.5 | 12.0 |
|   | Inactive | 22685 | 2065 | 2.3 | 83.2 | 14.3 |
| 3 | Active | 20782 | 2126 | 12.8 | 38.3 | 11.6 |
|   | Inactive | 21408 | 2202 | 2.3 | 81.8 | 14.1 |
| 4 | Active | 18339 | 2097 | 23.1 | 29.1 | 11.7 |
|   | Inactive | 20075 | 2092 | 2.1 | 80.9 | 14.1 |
| 5 | Active | 17927 | 2063 | 23.7 | 26.9 | 11.7 |
|   | Inactive | 20326 | 2068 | 2.9 | 79.5 | 14.1 |

**Notes.**
[a] Compliance and mean noncompliant speed for inactive SMA trips refer to trips with all speeds ≤10 knots and mean of all speeds >10 knots, respectively.

is conventional to consider the upper bound on the parameter's effect by estimating its maximum influence. A useful rule of thumb is to divide the parameter value by four to get an approximate upper bound on the effect. For example, the estimated median of the difference between active periods 2 and 1 for cargo ships is −0.02, which corresponds to a maximum drop of 0.09 in PDGT10 (from 0.50 to 0.41); by comparison, the median value of −1.07 for the difference between active periods 5 and 1 would take an expected PDGT10 value of 0.50 down to 0.16.

## RESULTS

A total of 201,862 trips made by 8,009 individual vessels were analyzed. In the first two active seasons of the speed restrictions (i.e., the regulated community's initial response to the novel regulation), 4.0% and 4.2% of the trips were fully compliant and PDGT10 values averaged 57.3% and 55.5%, respectively (Table 1; Fig. 2). In comparison, when speed restrictions were not in effect during the first two years of the regulations, 1.7% and 2.3% of the trips within the geographic boundaries of the SMAs were conducted entirely with speeds ≤10 knots and PDGT10 values were 83.4% and 83.2%, respectively (Table 1).

The largest response in PDGT10 values over time among the three vessel types analyzed was for cargo ships (Table 2). The temporal effect of the second active season relative to the first for this vessel class was significantly negative, with a median value of −0.02 (95% BCI [−0.06, 0.01]). This effect increased 35-fold for the third active season to −0.70 (−0.72, −0.66), dropped further in the fourth active season to −1.20 (−1.24, −1.17), and then to −1.07 (−1.11, −1.03) in the fifth active season. For tankers, there was a notable drop in expected PDGT10 beginning in the third active season, with the median seasonal difference dropping to −0.25 (−0.31, −0.18), and further to −0.48 (−0.54, −0.41) and −0.62 (−0.69, 0.56) in seasons four and five, respectively (Table 2; Fig. 3). The change in vessel speed for passenger vessels was less consistent, showing little change in the first

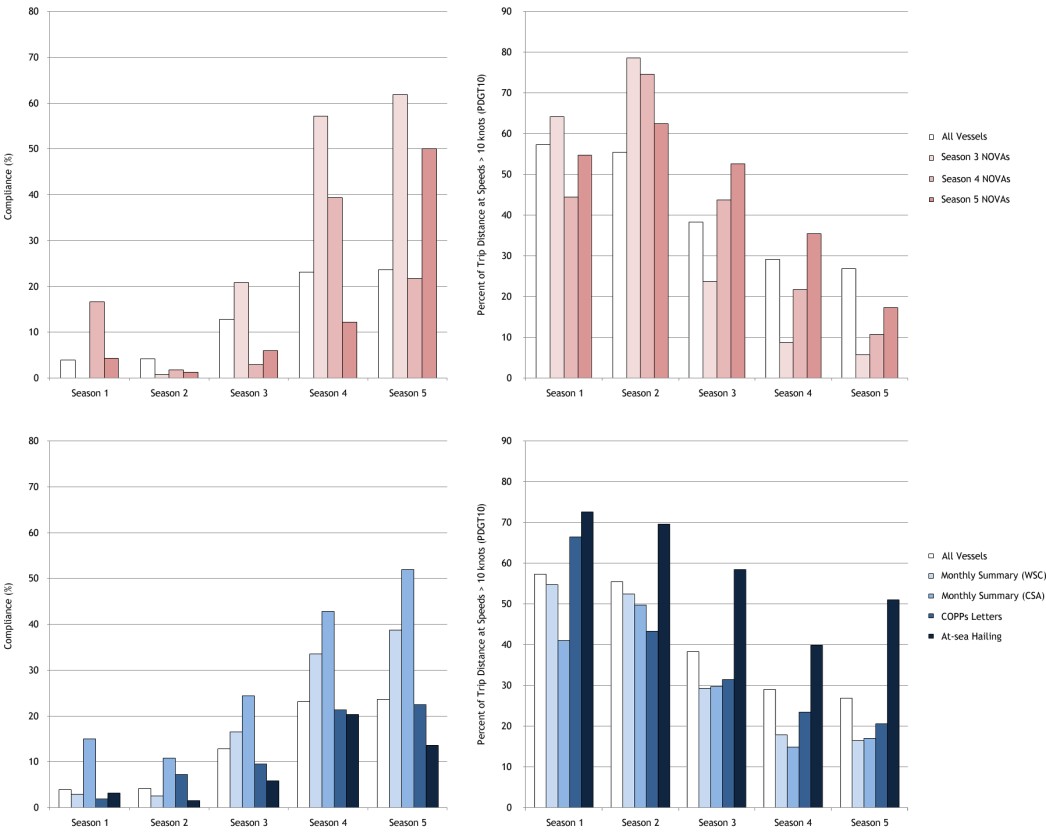

**Figure 2 Temporal changes in vessel speed restriction compliance metrics during the first five years of the regulations for vessels associated with the different notification/enforcement programs.** Compliance metrics for all vessels analyzed are also included for comparison and NOVA recipients have been further split based on when they received NOVAs (e.g. Season 3 NOVAs includes vessels that received their notices of violation shortly before or after the onset of Season 3) to better illustrate potential impacts associated with the enforcement action.

three active seasons (nominally higher in the third season) before becoming significantly negative in the fourth and fifth active season (Table 2; Fig. 3). None of the three models showed obvious lack of convergence, nor was there indication of lack of fit, based on the results of posterior predictive checks.

Of the notification programs studied, vessels hailed by the USCG seemingly exhibited the smallest relative change in compliance following their notification (Table 3); and, transits by this group subsequent to their notification were consistently higher than the population as a whole (Fig. 2). The average PDGT10 values of COPPS letter recipients decreased from 66.3% to 33.3% after being notified (Table 3), representing a clear but modest response to the program.

Vessels/companies that received NOVAs seemed to exhibit the greatest relative change in fully compliant trips and average PDGT10 after being cited. Average PDGT10 values went from 62.0% for trips prior to notification to 14.5% after fines were issued (Table 3). Average PDGT10 values for NOVA and monthly summary (both WSC and CSA) recipients declined in each successive active period following receipt of their respective

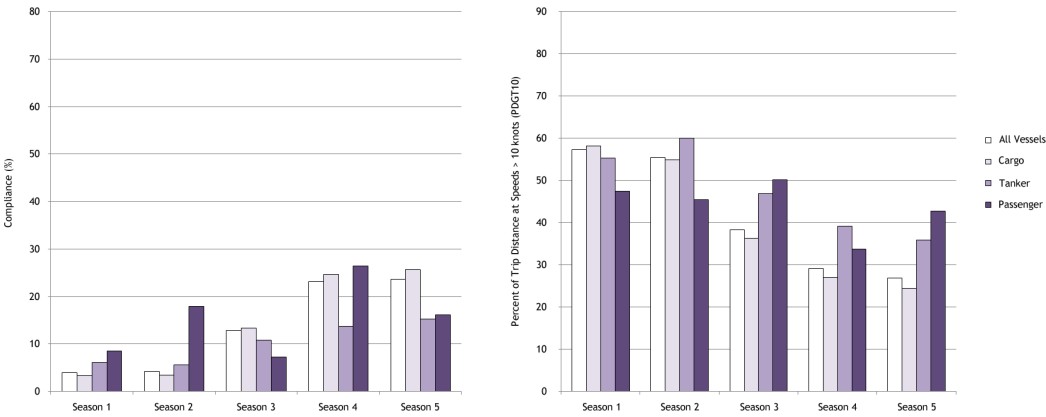

**Figure 3 Temporal changes in vessel speed restriction compliance metrics during the first five years of the regulations for the three principal vessel types analyzed.** Compliance metrics for all vessels analyzed are also included for comparison.

**Table 2 Model-based estimates of seasonal differences in PDGT10 for cargo, tanker and passenger vessels, along with posterior 95% credible intervals (highest posterior density intervals).** Each parameter represents the expected difference in PDGT10 in a specified season, relative to the first season, on the inverse-logit scale. Intervals that do not contain positive values are highlighted in bold.

| Vessel type | Season | Median | Standard deviation | 95% HPD interval |
|---|---|---|---|---|
| Cargo | 2 | −0.02 | 0.001 | (−0.06, 0.01) |
| | 3 | −0.70 | 0.001 | **(−0.74, −0.66)** |
| | 4 | −1.20 | 0.001 | **(−1.24, −1.17)** |
| | 5 | −1.07 | 0.001 | **(−1.11, −1.03)** |
| Tanker | 2 | 0.18 | 0.002 | (0.11, 0.25) |
| | 3 | −0.25 | 0.002 | **(−0.31, −0.18)** |
| | 4 | −0.48 | 0.002 | **(−0.54, −0.41)** |
| | 5 | −0.62 | 0.002 | **(−0.69, −0.56)** |
| Passenger | 2 | 0.12 | 0.008 | (−0.07, 0.32) |
| | 3 | 0.25 | 0.006 | (0.07, 0.41) |
| | 4 | −0.56 | 0.007 | **(−0.74, −0.39)** |
| | 5 | −0.48 | 0.007 | **(−0.65, −0.31)** |

enforcement/notification actions (Fig. 2). WSC monthly summary recipients made some of the largest relative adjustments in behavior (second only to NOVA recipients) with respect to full compliance (Table 3). Among the non-punitive programs, CSA monthly summary recipients had the greatest number of fully compliant trips (55.6%) and lowest average PDGT10 (12.3%) after being notified (Table 3).

## DISCUSSION

The U.S. Endangered Species Act (ESA) and related environmental legislation provide rather broad agency discretion for developing and implementing conservation regulations. However, without compliance, such regulations will be largely ineffective no matter how

**Table 3 Compliance metric summary statistics for trips through active SMAs by vessels associated with notification/enforcement programs both before and after the notification/enforcement.**

| Program | Timing | Trips | Vessels[a] | Compliance | PDGT10 | Mean noncompliant speed |
|---------|--------|-------|---------|------------|--------|-------------------------|
| At-sea hailing | Before | 964 | 46 | 4.9 | 70.3 | 13.2 |
| | After | 1260 | 44 | 11.8 | 48.7 | 12.1 |
| COPPs letter | Before | 1572 | 85 | 2.6 | 66.3 | 12.8 |
| | After | 2743 | 62 | 14.3 | 33.3 | 11.9 |
| Monthly summary (CSA) | Before | 2197 | 40 | 29.5 | 35.8 | 10.9 |
| | After | 2119 | 30 | 55.6 | 12.3 | 10.6 |
| Monthly summary (WSC) | Before | 14203 | 317 | 3.3 | 51.7 | 11.8 |
| | After | 19416 | 303 | 29.0 | 20.8 | 11.7 |
| NOVA | Before | 1318 | 28 | 3.3 | 62.0 | 13.0 |
| | After | 562 | 14 | 40.4 | 14.5 | 11.7 |

**Notes.**

[a] Not all vessels with trips prior to (or associated with) the initiation of their respective notification/enforcement program made subsequent trips through active SMAs.

well they are designed or how important their mandates are perceived. In our study, substantial modifications to normal practices were expected of a large, multi-national community to a novel ESA-promulgated regulation.

We found that, while much of the regulated community responded when vessel speed restrictions were instituted, a substantial number of trips were not in total compliance and the 10-knot limit was routinely exceeded. This suggests that extensive initial and ongoing efforts to inform the regulated community about the speed restrictions provided no assurances that widespread compliance would necessarily follow, even though this information was provided using virtually every available conventional maritime communications system and requirements that mariners fully understand applicable regulations while sailing in U.S. waters. In addition, non-punitive notifications to violators (i.e., radio contact at sea, COPPS letters) by recognized enforcement authorities resulted in only modest changes in compliance rates.

Due to the number and diversity of entities affected by this rule, it is possible that several years were needed for the community to incorporate speed limits into their operating procedures. It is worth noting, for example, that some printed and broadcast information about the restrictions may have become available to "foreign-flagged" vessels (a large portion of ships entering U.S. ports) primarily after entering U.S. waters. However, most commercial vessels studied here, including foreign-flagged vessels, are engaged in repeated, scheduled routes and likely were exposed multiple times each year to broadcast and broadly-disseminated information about the restrictions.

Our results indicate that in response to the restrictions vessel operators tended to use speeds that while not always less than 10 knots for the duration of a transit were nonetheless slower than they might otherwise use. At-sea speeds typically range from 10–15, 15–25, and 20–25 knots for tanker, cargo, and passenger vessels, respectively. Accordingly, cargo vessels, the most numerous vessel type in our study and the type most

named in enforcement actions, were required to make significant shifts in operations to comply with the speed regulation. Relative to cargo and passenger vessels, tankers needed to make the smallest changes in speeds to comply with the regulation, and it appears the approach taken by this vessel class was to reduce speeds when traveling through active SMAs (as reflected in their PDGT10 values), but, not to a point of full compliance.

The highest compliance rates were observed in the latter active periods, with notable changes occurring in the third season. Given the timing of the first set of NOVAs, these results suggest, but do not confirm, that the issuing of citations strongly influenced the behavior of notified vessels/companies. In addition, although they were issued to a fraction of the regulated community, citations appear to have improved compliance in the regulated population as a whole. This is consistent with findings by others whereby environmental monitoring and enforcement activities had a strong impact not only in reducing future violations (*Gray & Shimshack, 2011*), but also that deterrence resulting from these activities was almost as strong in affecting the compliance of others in the regulated community as it was on the sanctioned entity (*Shimshack & Ward, 2005*). Assessing internal business actions is beyond the scope of this study, but anecdotal reports to us indicate that there was broad knowledge among maritime industries that citations/fines were being issued. In addition, OLE press releases and industry trade publications notified readers about the issuing of fines and named the violator's company. Societal expectations, perceived social costs, and the importance of reputation have been identified as motivators in corporate compliance behavior (*May, 2005*; *Gunningham, Thornton & Kagan, 2005*), and these factors may have been at play in our study.

Each of the targeted notification programs appeared to have at least some effect on improved compliance in individual vessels. There are important distinctions between these programs that may be reflected in their relative effectiveness. An at-sea hailing incident may have been known only to the vessel operator and this program was limited geographically and temporally. Its modest influence on compliance suggests that when the perceived likelihood of detection is low (no visible enforcement entity present on the majority of trips) the threat of adverse consequences is also low. Receipt of NOVAs or monthly summaries of operations to association members (and perhaps COPPS letters) was almost certainly known throughout a given company (in most cases, company officials were the entities being notified) which may have led to company-wide directives regarding compliance. CSA members comprise a diverse set of vessel types, tankers being strongly represented; likely, minimal alteration of operations was needed for many of these vessels to comply. In addition, many CSA-member vessels are engaged largely in domestic trade and in making repeated U.S. port entries may have been exposed to a greater degree than other vessels to awareness-raising about the restrictions.

Multiple notification/enforcement programs can have an additive value in influencing compliance rates (*Gray & Shimshack, 2011*) and the threat of punitive actions may bolster the effectiveness of non-punitive measures (*Abbot , 2009*; *Scholz & Gray, 1990*). We note that shortly after NOVAs were issued the industry associations sought to develop regular non-compliance notification programs for their members. Therefore, these follow-up

programs likely complimented enforcement actions and provided periodic reminders that operations were being routinely monitored.

Enforcement activities can be labor- and resource-intensive and may be difficult if the regulated population is large or widely dispersed (*Abbot , 2009*; *Ali & Abdullah, 2010*). Where feasible, remote-monitoring can be a cost-effective means to improve compliance (*Purdy, 2010*). Whereas we did not attempt to quantify agency costs involved in the monitoring/enforcement activities described here, by utilizing an existing infrastructure for remote monitoring and relying on electronic means or surface mail for nearly all enforcement and notification activities, costs were almost certainly considerably less than those involved in conventional inspection or law enforcement activities.

The vessel speed restrictions appear to be working as intended: no fatal vessel strike-related right whale deaths were reported in or near active SMAs since the rule went into effect, a period that is nearly twice the longest interval between subsequent known vessel collision fatalities in these same areas in an 18-year study period prior to adoption of the rule (*Laist, Knowlton & Pendleton, 2014*). Modeling studies have indicated that the risk of fatal vessel collisions of right whales has been reduced by the vessel speed restriction (*Lagueux et al., 2011*; *Wiley et al., 2011*). The probability (a 80–90% reduction in risk) of fatal vessel collisions was lowest in the latter part of the period in which the rule was in effect (*Conn & Silber, 2013*), during which improved compliance rates were observed.

Voluntary actions and incentives are approaches that have been widely used and can be effective in reducing environmental impacts (*Dietz & Stern, 2002*; *Gunningham, Kagan & Thornton, 2004*; *Stafford, 2012*). However, in regard to the conservation issue of vessel strikes of large whales, mandatory and enforced changes in vessel operations appear to have considerable conservation value while adherence to—and therefore effectiveness of—previously implemented voluntary measures to reduce whale disturbance (*Wiley et al., 2008*) and vessel/whale collisions (*Silber, Adams & Bettridge, 2012*) was low.

Costs incurred in issuing and enforcing living resource conservation regulations and costs to regulated entities might be assessed relative to societal benefits (*Gren & Li, 2011*). Economic impacts to the regulated community arising from vessel speed restrictions (including the effect of lost time, indirect impacts to intermodal transport systems etc.) are a fraction of the value of the goods and services provided by the affected maritime and associated industries (*Nathan Associates Inc., 2012*), and these might be weighed in the context of societal valuation studies of the virtues of preserving endangered and threatened species (e.g., *Wallmo & Lew, 2011*).

## SUMMARY AND CONCLUSIONS

This study provides information about the relative roles of punitive and non-punitive targeted actions designed to enhance compliance. Our findings, like those of others, appear to strongly suggest that citations/fines were motivators in improving compliant behavior and these may have been backed by targeted notifications of violation. Progressively improving compliance rates appeared to have been influenced, to varying degrees, by broad-scale notification programs and the threat (or reality) of enforcement activities. These results may

help in formulating management strategies for this particular conservation concern and in improving compliance in virtually any setting in which regulatory compliance is sought.

## ACKNOWLEDGEMENTS

K Chin and D Phinney of the John A. Volpe National Transportation Systems Center have been invaluable to acquiring and analyzing AIS data. The USCG National AIS program has been vital to this study. We thank the WSC and CSA for making their members aware of NOAA's program to share compliance data with vessel owners, operators, and charterers. Assistance in various forms and comments provided by S Bettridge, B Sousa, J Landon, and D Smith improved the manuscript. One of us (GKS) dedicates this work to Robert L. Silber whose lifetime of innovation and compassion in service to others have provided constant inspiration.

### Funding

Funding data acquisition and analysis was provided by the Office of Protected Resources. The funders had no role in study design, data collection and analysis, decision to publish, or preparation of the manuscript.

### Grant Disclosures

The following grant information was disclosed by the authors:
Office of Protected Resources.

### Competing Interests

The authors declare there are no competing interests.

### Author Contributions

- Gregory K. Silber conceived and designed the experiments, wrote the paper, reviewed drafts of the paper, principal role in organizing the project and in securing data for analysis.
- Jeffrey D. Adams performed the experiments, analyzed the data, contributed reagents/materials/analysis tools, wrote the paper, prepared figures and/or tables, reviewed drafts of the paper.
- Christopher J. Fonnesbeck performed the experiments, analyzed the data, contributed reagents/materials/analysis tools, wrote the paper, reviewed drafts of the paper.

### Supplemental information

Supplemental information for this article can be found online at http://dx.doi.org/10.7717/peerj.399.

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
