# Peer review of "Compliance with vessel speed restrictions to protect North Atlantic right whales"

_PeerJ, doi:10.7717/peerj.399_

## Round 0.1 · original submission · Minor Revisions

The manuscript is clear and concise, but as noted by one of the reviewers there is an odd transition in the text when describing the compliance/enforcement programs that should be addressed - perhaps by a more inclusive description in the introducing paragraph) There is another odd transition at page 276. It may be that the following paragraph is out of order?

After addressing the changes provided in the reviews, and in the PDF returned by reviewer number 2, this paper should be acceptable for publication.

·

Basic reporting

The article is clearly written, easy to follow, and the discussion and conclusions are well supported by the data in the study.

My only minor suggestions for editing include a strange break in the text between line 80/81 where there is an abrupt and unclear transition, accompanied by a change of font. It was unclear whether text was missing, or whether a transition session could be added at this point to improve clarity. Figure 3 seems unnecessary given the data presented in the earlier figures and tables.

Experimental design

No comments.

Validity of the findings

No comments.

Additional comments

This study reports on the remote monitoring of compliance to a speed reduction rule imposed for protection of an endangered species of whale. The authors present a compelling study that indicates that enforcement, and financial punitive measures in particular, appeared to play a large role in compliance of this new regulation. The study highlights the need to consider and outline enforcement efforts when new environmental regulations are put into place. The article is clearly written, easy to follow, and the discussion and conclusions are well supported by the data in the study.

·

Basic reporting

No comments

Experimental design

The experimental design seems appropriate. Since I am not a Bayesian mathmetician I did not feel qualified to review all of the Bayesian approaches although the logic of the approach made sense to me. As I noted in the text I had a concern in Table 2 about why the # of vessels in the before and after groups (since the way I read it the change in a given vessel was being measured before/after) were different and suggest it either be rephrased in the text so the reader understands why these #'s would be different.

Validity of the findings

No comments

Additional comments

I feel this is a great paper with a well-thought out approach to understanding the degree of compliance with this regulation. I made a variety of minor comments on the attached pdf. The figures seemed to be out of order and the captions didn't always seem to be correct for the given figure so this needs to be checked carefully. Also, I noticed a couple of references were missing or not mentioned in the text and I think should be but I did not check this aspect carefully. But overall I think the authors have done a great job with this complex issue.

Reviewer 3 ·

Basic reporting

I was asked to comment specifically on the statistics. They appear sound. The analysis is simple and straightforward, the modelling approach is well described, and the authors justified some of the few non-standard / system specific steps they took to prepare and analyze the vessel data.

If anything, the approach is perhaps a little overboard, in that this model probably could have been fit using any number of R packages. That said, using a Bayesian MCMC approach should work just fine.

Experimental design

This was an analytical project that took advantage of vessel monitoring data, it was not experimental in nature. The analysis of vessel compliance to speed-limit regulations in right whale areas is sound, interesting, and of management interest.

Validity of the findings

I am convinced the authors have conducted their statistical analysis properly and that the results they report are sound.

---

## Round 0.2 · accepted · Accept

I wholeheartedly agree with the reviewers final guidance. Thanks for the rapid response to the requested revisions.

·

Basic reporting

Revisions fine

Experimental design

Revisions fine

Validity of the findings

Revisions fine

Additional comments

All of the edits look good to me and the paper seems ready to go.